## [Decision Letter · Decision Letter 0]

7 Aug 2025

Dear Dr. Kalu,

Thank you for submitting your manuscript to PLOS ONE. After careful consideration, we feel that it has merit but does not fully meet PLOS ONE’s publication criteria as it currently stands. Therefore, we invite you to submit a revised version of the manuscript that addresses the points raised during the review process.

We look forward to receiving your revised manuscript.

Kind regards,

Md. Asaduzzaman, Ph.D., M. Engg.

Academic Editor

PLOS ONE

Journal Requirements:

5. Please ensure that you refer to Figure 1 in your text as, if accepted, production will need this reference to link the reader to the figure.

6. Please upload a copy of Figure 4, to which you refer in your text on page 23 in PDF submission. If the figure is no longer to be included as part of the submission please remove all reference to it within the text.

7. We note that Figure 1 in your submission contain [map/satellite] images which may be copyrighted. All PLOS content is published under the Creative Commons Attribution License (CC BY 4.0), which means that the manuscript, images, and Supporting Information files will be freely available online, and any third party is permitted to access, download, copy, distribute, and use these materials in any way, even commercially, with proper attribution. For these reasons, we cannot publish previously copyrighted maps or satellite images created using proprietary data, such as Google software (Google Maps, Street View, and Earth). For more information, see our copyright guidelines: http://journals.plos.org/plosone/s/licenses-and-copyright.

Reviewers' comments:

Reviewer's Responses to Questions

**Comments to the Author**

1. Is the manuscript technically sound, and do the data support the conclusions?

Reviewer #1: Yes

Reviewer #2: Yes

2. Has the statistical analysis been performed appropriately and rigorously?

Reviewer #1: Yes

Reviewer #2: Yes

3. Have the authors made all data underlying the findings in their manuscript fully available?

Reviewer #1: Yes

Reviewer #2: No

4. Is the manuscript presented in an intelligible fashion and written in standard English?

Reviewer #1: Yes

Reviewer #2: Yes

Reviewer #1: On account of the manuscript PONE-D-25-40356, entitled “Physicochemical status of Nwanedi River water, and its influence on the metabolome of river-irrigated tomato leaves” by Leornard Ntanganedzeni Musweswe et al., the authors evaluated the physicochemical status, pH, total dissolved solids (TDS), electrical conductivity (EC), salinity (SAL), and metals of Nwanedi River water in South Africa and its influence on the primary metabolome of disordered tomato leaves irrigated with the river water. The topic is important to better understanding of the associations between irrigation water status and the primary metabolome of leaves irrigated with the water, and environmental management of Nwanedi River as well. After careful consideration, I feel that this manuscript is to be published after improvement of some minor shortcomings. Details of my comments are as follows:

The manuscript was well written and designed, and the authors got interesting results. Only minor revisions are required before publication. Although the authors analyzed metal and metabolome of disordered tomato leaves, experimental validation of quantification such as recovery rate, matrix effect, the calibration curve with linearity for quantification, and limit of detection (LOD) and limit of quantification (LOQ) is missing in the present manuscript. The authors are encouraged to show these results in the manuscript for enhancement of the accuracy and reliability of the results. After that I am ready to recommend the present manuscript for publication.

Reviewer #2: The manuscript titled "Physicochemical status of Nwanedi River water, and its influence on the metabolome of river-irrigated tomato leaves" presents a comprehensive investigation into the impact of irrigation water quality on tomato plant metabolism. The study is well-structured and addresses a significant gap in understanding how the physicochemical parameters of river water influence plant health. However, several areas require clarification and improvement to enhance the manuscript's rigor and readability.

1. Abstract and Focus

1. The abstract is informative but could be more concise. Consider streamlining the methodology and results sections to focus on key findings.

2. Clarify the term "disordered tomato leaves" — is this a specific physiological condition or a general term for stressed leaves?

2. Introduction and Objectives 3. The introduction provides a good background but could better highlight the study's novelty. Emphasize why this specific river and crop combination is unique or understudied. 4. Include a clearer hypothesis or research objectives upfront to guide the reader.

3. Methodology Improvements 5. Sampling: Provide more details on the selection criteria for "disordered" leaves. How was disorder quantified or identified? 6. Metabolite Analysis: Specify the rationale for selecting the 25 primary metabolites analyzed. Were these chosen based on prior literature or preliminary screening? 7. Statistical Analysis: Clarify how the Pearson correlation analysis accounts for potential confounding variables (e.g., soil properties, climate).

4. Results Presentation 8. Table 3 is cut off in the provided document. Ensure that all tables are complete and clearly labeled. 9. Figure 2 (heatmap) is described but not included in the text. Verify its placement and provide a clearer interpretation of clustering patterns. 10. Physicochemical Data: Highlight whether spatial trends in water quality (e.g., downstream vs. upstream) were observed, as this could inform mitigation strategies.

5. Discussion Enhancement 11. The discussion is thorough but somewhat repetitive. Condense the interpretation of correlations (e.g., hypoxanthine) and focus on mechanistic insights. 12. Compare findings more directly with similar studies. For example, how do the observed metabolite changes align with known stress responses in tomatoes or other crops? 13. Address the practical implications: What do the results suggest for farmers using Nwanedi River water? Are there immediate risks or long-term adaptation strategies?

6. Conclusion and Future Directions 14. The conclusion could better summarize the study's broader significance. For example, how might these findings inform water quality policies or agricultural practices in the region? 15. Mention limitations (e.g., single-season sampling, lack of soil data) and suggest future research directions (e.g., multi-year studies, transcriptomics).

7. References and Literature 16. Ensure all citations are formatted consistently per PLOS ONE guidelines. 17. Include recent literature (post-2020) where relevant to support claims about metabolite roles in stress responses.

8. Data Availability and Technical Issues 18. The manuscript states data are fully available, but does not provide a repository link or accession number. Clarify where and how data can be accessed. 19. Define abbreviations (e.g., SAL, TDS) at first use. 20. Check for grammatical errors (e.g., "disordered leaves" vs. "stressed leaves"). 21. Simplify technical jargon in places (e.g., "marked influence" could be "significant impact").

**Do you want your identity to be public for this peer review?** For information about this choice, including consent withdrawal, please see our Privacy Policy

Reviewer #1: No

Reviewer #2: **Yes:** Jonah Bawa Adokwe Ph.D

---

## [Author Response · Author response to Decision Letter 1]

27 Oct 2025

The reviewers and editor did well. I appreciate their work.

---

## [Decision Letter · Decision Letter 1]

9 Dec 2025

Dear Dr. Kalu,

We look forward to receiving your revised manuscript.

Kind regards,

Ying Ma, Ph.D.

Academic Editor

PLOS One

Journal Requirements:

Reviewers' comments:

Reviewer's Responses to Questions

**Comments to the Author**

Reviewer #1: All comments have been addressed

Reviewer #2: (No Response)

2. Is the manuscript technically sound, and do the data support the conclusions?

Reviewer #1: Yes

Reviewer #2: Partly

3. Has the statistical analysis been performed appropriately and rigorously?

Reviewer #1: Yes

Reviewer #2: No

4. Have the authors made all data underlying the findings in their manuscript fully available?

Reviewer #1: Yes

Reviewer #2: Yes

5. Is the manuscript presented in an intelligible fashion and written in standard English?

Reviewer #1: Yes

Reviewer #2: Yes

Reviewer #1: On account of the manuscript PONE-D-25-40356R1, entitled “Physicochemical status of Nwanedi River water, and its influence on the metabolome of river-irrigated tomato leaves” by Leornard Ntanganedzeni Musweswe et al., the authors the authors revised the manuscript appropriately according to the Reviewers comments. After careful consideration, the present manuscript is ready for acceptance for publication.

Reviewer #2: The authors have done a good job in dealing with the first concerns. The paper is significantly better, especially in explaining the novelty and practical implications of the study. Nevertheless, there are several methodological and data reporting concerns that should be addressed before it can be recommended for publication:

Comments:

1. Table 4 indicates linearity (r 2 =) = 1.000 at all 13 analytes. This is statistically unlikely and indicates the problem of rounding or the reporting of data. Please give the actual r 2 values to at least 3 decimal places (e.g., 0.998, 0.9995).

2. The study would use some discussion of sample size limitations, since using the number of water sampling points is the main statistical method (only 5 points), and correlation analysis is the main statistical method. Recognition of the fact that correlations do not imply causation. Understanding that there were confounding variables (soil properties, microclimate) that were not directly measured.

3. The reason why it does not give information on recovery rates is unsatisfactory. Although the mentioned references utilize more or less the same instrumentation, the extraction efficiency as well as the analytical accuracy still significant to be validated concerning the metabolomics studies. Please explain: Have there been any control samples? How was the precision of the method determined? What were the quality control measures used?

4. Table 5: The difference in retention time of the samples needs to be presented with precise measures (RSD) to indicate the reliability of the method.

5. Grammatical errors that had to be fixed:

• Line 157: "10mg" → "10 mg"

• Line 452: "Move over." → "Moreover."

• Fig 3 caption: "fig 3" → "Fig 3."

6. The conclusion provides future directions, but it does not pay enough attention to the present limitations of the study. Single-season sampling. Please specify the discussion of: Single-season sampling. The confounding factors (particularly the soil properties) are not measurable. Limitations on the sample size used to test correlation.

It will add value to the literature once the above issues have been clarified or corrected. My suggestion is a slight change prior to acceptance.

**Do you want your identity to be public for this peer review?** For information about this choice, including consent withdrawal, please see our Privacy Policy

Reviewer #1: No

Reviewer #2: **Yes:** Dr. Jonah Bawa Adokwe

---

## [Decision Letter · Decision Letter 2]

25 Jan 2026

Physicochemical Status of Nwanedi River Water, and its Influence on the Metabolome of River-Irrigated Tomato Leaves

PONE-D-25-40356R2

Dear Dr. Kalu,

We’re pleased to inform you that your manuscript has been judged scientifically suitable for publication and will be formally accepted for publication once it meets all outstanding technical requirements.

Kind regards,

Ying Ma, Ph.D.

Academic Editor

PLOS One

Additional Editor Comments (optional):

Reviewers' comments:

Reviewer's Responses to Questions

**Comments to the Author**

Reviewer #2: All comments have been addressed

2. Is the manuscript technically sound, and do the data support the conclusions?

Reviewer #2: Yes

3. Has the statistical analysis been performed appropriately and rigorously?

Reviewer #2: Yes

4. Have the authors made all data underlying the findings in their manuscript fully available?

Reviewer #2: Yes

5. Is the manuscript presented in an intelligible fashion and written in standard English?

Reviewer #2: Yes

Reviewer #2: The authors have satisfactorily addressed all comments from the previous review round. The manuscript now includes proper quality control documentation, realistic linearity values, retention time variability data, and an appropriate discussion of study limitations. I recommend acceptance for publication

**Do you want your identity to be public for this peer review?** For information about this choice, including consent withdrawal, please see our Privacy Policy

Reviewer #2: **Yes:** Dr. Jonah Bawa Adokwe

---

## [Editor Report · Acceptance letter]

PONE-D-25-40356R2

PLOS One

Dear Dr. Kalu,

I'm pleased to inform you that your manuscript has been deemed suitable for publication in PLOS One. Congratulations! Your manuscript is now being handed over to our production team.

Kind regards,

on behalf of

Dr. Ying Ma

Academic Editor

PLOS One